# Towards neural networks that provably know when they don't know

**Alexander Meinke**
University of Tübingen

**Matthias Hein**
University of Tübingen

## Abstract

It has recently been shown that ReLU networks produce arbitrarily over-confident predictions far away from the training data. Thus, ReLU networks do not know when they don't know. However, this is a highly important property in safety critical applications. In the context of out-of-distribution detection (OOD) there have been a number of proposals to mitigate this problem but none of them are able to make any mathematical guarantees. In this paper we propose a new approach to OOD which overcomes both problems. Our approach can be used with ReLU networks and provides provably low confidence predictions far away from the training data as well as the first certificates for low confidence predictions in a neighborhood of an out-distribution point. In the experiments we show that state-of-the-art methods fail in this worst-case setting whereas our model can guarantee its performance while retaining state-of-the-art OOD performance.[1]

## 1 Introduction

Deep Learning Models are being deployed in a growing number of applications. As these include more and more systems where safety is a concern, it is important to guarantee that deep learning models work as one expects them to. One topic that has received a lot of attention in this area is the problem of adversarial examples, in which a model's prediction can be changed by introducing a small perturbation to an originally correctly classified sample. Achieving robustness against this type of perturbation is an active field of research. Empirically, adversarial training (Madry et al., 2018) performs well and provably robust models have been developed (Hein & Andriushchenko, 2017; Wong & Kolter, 2018; Raghunathan et al., 2018; Mirman et al., 2018; Cohen et al., 2019).

On the other end of the spectrum it is also important to study how deep learning models behave far away from the training samples. A simple property every classifier should satisfy is that far away from the training data, it should yield close to uniform confidence over the classes: it knows when it does not know. However, several cases of high confidence predictions far away from the training data have been reported for neural networks, e.g. fooling images (Nguyen et al., 2015), for out-of-distribution (OOD) images (Hendrycks & Gimpel, 2017a) or in medical diagnosis (Leibig et al., 2017). Moreover, it has been observed that, even on the original task, neural networks often produce overconfident predictions (Guo et al., 2017). Very recently, it has been shown theoretically that the class of ReLU networks (all neural networks which use a piecewise affine activation function), which encompasses almost all standard models, produces predictions with arbitrarily high confidences far away from the training data (Hein et al., 2019). Unfortunately, this statement holds for almost all such networks and thus without a change in the architecture one cannot avoid this phenomenon.

Traditionally, the calibration of the confidence of predictions has been considered on the in-distribution (Guo et al., 2017; Lakshminarayanan et al., 2017a). However these techniques cannot be used for detection (Leibig et al., 2017). Only recently the detection of OOD inputs (Hendrycks & Gimpel, 2017a) has been tackled. The existing approaches are roughly of two types: first, postprocessing techniques that adjust the estimated confidence (DeVries & Taylor, 2018; Liang et al., 2018) which includes the baseline ODIN. Second, modification of the classifier training by integrating generative models like a VAE or GAN in order to discriminate out-distribution from in-distribution data (Lee et al., 2018a; Wang et al., 2018; Lee et al., 2018b) or approaches which enforce low

---

[1]Code at `https://github.com/AlexMeinke/certified-certain-uncertainty`

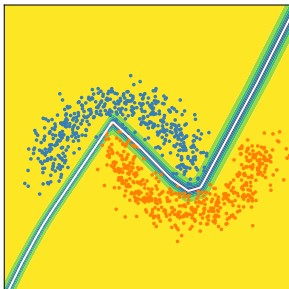 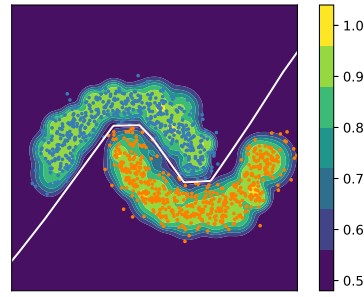

Figure 1: **Illustration on toy dataset:** We show the color-coded confidence in the prediction (yellow indicates high confidence $\max_y \hat{p}(y|x) \approx 1$, whereas dark purple regions indicate low confidence $\max_y \hat{p}(y|x) \approx 0.5$) for a normal neural network (left) and our CCU neural network (right). The decision boundary is shown in white which is similar for both models. Our CCU-model retains high-confidence predictions in regions close to the training data, whereas far away from the training the CCU-model outputs close to uniform confidence. In contrast the normal neural network is over-confident everywhere except very close to the decision boundary.

confidence on OOD inputs during training (Hein et al., 2019; Hendrycks et al., 2019). Worst-case aspects of OOD detection have previously been studied in Nguyen et al. (2015); Schott et al. (2018); Hein et al. (2019); Sehwag et al. (2019), but no robustness guarantees have yet been proposed for this setting. A generalization guarantee for an out-of-distribution detection scheme is provided in Liu et al. (2018). While this is the only guarantee we are aware of, it is quite different from the type of guarantees we present in this paper. In particular, none of those approaches are able to guarantee that neural networks produce low confidence predictions far away from the training data. We prove that our classifier satisfies this requirement even when we use ReLU networks as the classifier model - without loosing performance on either the prediction task on the in-distribution nor the OOD detection performance, see Figure 1 for an illustration. Moreover, our technique allows to give upper bounds on the confidence over a whole neighborhood around a point (worst-case guarantees). We show that most state-of-the-art OOD methods can be fooled by maximizing the confidence in this ball even when starting from uniform noise images, which should be trivial to identify. The central difference from existing OOD-methods is that we have a Bayesian framework for in-and out-distribution, where we model in-and out-distribution separately. In this framework our algorithm for training neural networks follows directly as maximum likelihood estimator which is different from the more ad-hoc methods proposed in the literature. The usage of Gaussian mixture models as the density estimator is then the essential key to get the desired provable guarantees.

## 2 A GENERIC MODEL FOR CLASSIFIERS WITH CERTIFIED LOW CONFIDENCE FAR AWAY FROM THE TRAINING DATA

The model which we propose in this paper assumes that samples from an out-distribution are given to us. In image recognition we could either see the set of all images as a sample from the out-distribution (Hendrycks et al., 2019) or consider the agnostic case where we use use uniform noise on $[0, 1]^d$ as a maximally uninformative out-distribution. In both settings one tries to discriminate these out-distribution images from images coming from a particular image recognition task and the task is to get low confidence predictions on the out-distribution images vs. higher confidence on the images from the actual task. From the general model we derive under minimal assumptions a maximum-likelihood approach where one trains both a classifier for the actual task and density estimators for in- and out-distribution jointly. As all of these quantities are coupled in our model for the conditional distribution $p(y|x)$ we get guarantees by controlling the density estimates far away from the training data. This is a crucial difference to the approaches of Lee et al. (2018a); Wang et al. (2018); Hendrycks et al. (2019) which empirically yield good OOD performance but are not able to certify the detection mechanism.

### 2.1 A PROBABILISTIC MODEL FOR IN- AND OUT-DISTRIBUTION DATA

We assume that there exists a joint probability distribution $p(y, x)$ over the in- and out-distribution data, where $y$ are the labels in $\{1, \ldots, M\}$, $M$ is the number of classes, and $x \in \mathbb{R}^d$, where $d$ is the input dimension. In the following, we denote the underlying probabilities/densities with $p(y|x)$ resp.

$p(x)$ and the estimated quantities with $\hat{p}(y|x)$ and $\hat{p}(x)$. We are mainly interested in a discriminative framework, i.e. we want to estimate $p(y|x)$ which one can represent via the conditional distribution of the in-distribution $p(y|x,i)$ and out-distribution $p(y|x,o)$:

$$p(y|x) = p(y|x,i)p(i|x) + p(y|x,o)p(o|x) = \frac{p(y|x,i)p(x|i)p(i) + p(y|x,o)p(x|o)p(o)}{p(x|i)p(i) + p(x|o)p(o)}. \quad (1)$$

Note that at first it might seem strange to have a conditional distribution $p(y|x,o)$ for out-distribution data, but until now we have made no assumptions about what in-and out-distribution are. A realistic scenario would be that at test time we are presented with instances $x$ from other classes (out-distribution) for which we expect a close to uniform $p(y|x,o)$.

Our model for $\hat{p}(y|x)$ has the same form as $p(y|x)$

$$\hat{p}(y|x) = \frac{\hat{p}(y|x,i)\hat{p}(x|i)\hat{p}(i) + \hat{p}(y|x,o)\hat{p}(x|o)\hat{p}(o)}{\hat{p}(x|i)\hat{p}(i) + \hat{p}(x|o)\hat{p}(o)}. \quad (2)$$

Typically, out-distribution data has no relation to the actual task and thus we would like to have uniform confidence over the classes. Therefore we set in our model

$$\hat{p}(y|x,o) = \frac{1}{M} \quad \text{and} \quad \hat{p}(y|x,i) = \frac{e^{f_y(x)}}{\sum_{k=1}^{M} e^{f_k(x)}}, \quad y \in \{1, \dots M\}, \quad (3)$$

where $f : \mathbb{R}^d \to \mathbb{R}^M$ is the classifier function (logits). This framework is generic for classifiers trained with the cross-entropy (CE) loss (as the softmax function is the correct link function for the CE loss) and we focus in particular on neural networks. For a ReLU network the classifier function $f$ is componentwise a continuous piecewise affine function and has been shown to produce asymptotically arbitrarily highly confident predictions (Hein et al., 2019), i.e. the classifier gets more confident in its predictions the further it moves away from its training data. One of the main goals of our proposal is to fix this behavior of neural networks in a provable way.

Note that with the choice of $\hat{p}(y|x,o)$ and non-zero priors for $\hat{p}(i), \hat{p}(o)$, the full model $\hat{p}(y|x)$ can be seen as a calibrated version of $\hat{p}(y|x,i)$, where $\hat{p}(y|x) \approx \hat{p}(y|x,i)$ for inputs with $\hat{p}(x|i) \gg \hat{p}(x|o)$ and $\hat{p}(y|x) \approx \frac{1}{M}$ if $\hat{p}(x|i) \ll \hat{p}(x|o)$. However, note that only the confidence in the prediction $\hat{p}(y|x)$ is affected, the classifier decision is still done according to $\hat{p}(y|x,i)$ as the calibration does not change the ranking. Thus even if the OOD data came from the classification task we would like to solve, the trained classifier's performance would be unaffected, only the confidence in the prediction would be damped.

For the marginal out-distribution $\hat{p}(x|o)$ there are two possible scenarios. In the first case one could concentrate on the worst case where we assume that $p(x|o)$ is maximally uniformative (maximal entropy). This means that $\hat{p}(x|o)$ is uniform for bounded domains e.g. for images which are in $[0,1]^d$, $\hat{p}(x|o) = 1$ for all $x \in [0,1]^d$, or $\hat{p}(x|o)$ is a Gaussian for the domain of $\mathbb{R}^d$ (the Gaussian has maximum entropy among all distributions of fixed variance). However, in this work we follow the approach of Hendrycks et al. (2019) where they used the 80 million tiny image dataset (Torralba et al., 2008) as a proxy of all possible images. Thus we estimate the density of $\hat{p}(x|o)$ using this data.

In order to get guarantees, the employed generative models for $\hat{p}(x|i)$ and $\hat{p}(x|o)$ have to be chosen in a way that allows one to control predictions far away from the training data. Variational autoencoders (VAEs) (Kingma & Welling, 2014; Rezende et al., 2014), normalizing flows (Dinh et al., 2016; Kingma & Dhariwal, 2018) and generative adversarial networks (GANs) (Goodfellow et al., 2014) are powerful generative models. However, there is no direct way to control the likelihood far away from the training data. Moreover, it has recently been discovered that VAEs, flows and GANs also suffer from overconfident likelihoods (Nalisnick et al., 2019; Hendrycks et al., 2019) far away from the data they are supposed to model as well as adversarial samples (Kos et al., 2017).

For $\hat{p}(x|o)$ and $\hat{p}(x|i)$ we use a Gaussian mixture model (GMM) which is less powerful than a VAE but has the advantage that the density estimates can be controlled far away from the training data:

$$\hat{p}(x|i) = \sum_{k=0}^{K_i} \alpha_k \exp\left(-\frac{d(x,\mu_k)^2}{2\sigma_k^2}\right), \qquad \hat{p}(x|o) = \sum_{l=0}^{K_o} \beta_l \exp\left(-\frac{d(x,\nu_l)^2}{2\theta_l^2}\right) \quad (4)$$

where $K_i, K_o \in \mathbb{N}$ are the number of centroids and $d : \mathbb{R}^d \times \mathbb{R}^d \to \mathbb{R}$ is the metric

$$d(x,y) = \left\| C^{-\frac{1}{2}}(x-y) \right\|_2,$$

with $C$ being a positive definite matrix and

$$\alpha_k = \frac{1}{K_i} \frac{1}{(2\pi\sigma_k^2 \det C)^{\frac{d}{2}}}, \quad \beta_l = \frac{1}{K_o} \frac{1}{(2\pi\theta_l^2 \det C)^{\frac{d}{2}}}.$$

We later fix $C$ as a slightly modified covariance matrix of the in-distribution data (see Section 4 for details). Thus one just has to estimate the centroids $\mu_k, \nu_l$ and the variances $\sigma_k^2, \theta_l^2$. The idea of this metric is to use distances adapted to the data-distribution. Note that equation 4 is a properly normalized density in $\mathbb{R}^d$.

## 2.2 Maximum likelihood estimation

Given models for $\hat{p}(y|x)$ and $\hat{p}(x)$ we effectively have a full generative model and apply maximum likelihood estimation to get the underlying classifier $\hat{p}(y|x, i)$ and the parameters of the Gaussian mixture models $\hat{p}(x|i), \hat{p}(x|o)$. The only free parameter left is the probability $\hat{p}(i), \hat{p}(o)$ which we write compactly as $\lambda = \frac{\hat{p}(o)}{\hat{p}(i)}$. In principle this parameter should be set considering the potential cost of over-confident predictions. In our experiments we simply fix it to $\lambda = 1$.

$$\underset{(x,y)\sim p(x,y)}{\mathbb{E}} \log\left(\hat{p}(y,x)\right) = \underset{(x,y)\sim p(x,y)}{\mathbb{E}} \log\left(\hat{p}(y|x)\right) + \log(\hat{p}(x)),$$

$$= \underset{(x,y)\sim p(x,y)}{\mathbb{E}} \log\left(\frac{\hat{p}(y|x,i)\hat{p}(x|i)\hat{p}(i) + \frac{1}{M}\hat{p}(x|o)\hat{p}(o)}{\hat{p}(x|i)\hat{p}(i) + \hat{p}(x|o)\hat{p}(o)}\right) + \log\left(\hat{p}(x|i)\hat{p}(i) + \hat{p}(x|o)\hat{p}(o)\right). \quad (5)$$

In practice, we have to compute empirical expectations from finite training data from the in-distribution $(x_i, y_i)_{i=1}^{n_i}$ and out-distribution $(z_j)_{j=1}^{n_o}$. Labels for the out-distribution could be generated randomly via $p(y|x, o) = \frac{1}{M}$, but we obtain an unbiased estimator with lower variance by averaging over all classes directly, as was done in Lee et al. (2018a); Hein et al. (2019); Hendrycks et al. (2019). Now we can estimate the classifier $f$ and the mixture model parameters $\mu, \nu, \sigma, \theta$ via

$$\underset{f,\mu,\nu,\sigma,\theta}{\arg\max} \left\{ \frac{1}{n_i} \sum_{i=1}^{n_i} \log\left(\hat{p}(y_i|x_i)\right) + \frac{\lambda}{n_o} \sum_{j=1}^{n_o} \frac{1}{M} \sum_{m=1}^{M} \log\left(\hat{p}(m|z_j)\right) \right.$$

$$\left. + \frac{1}{n_i} \sum_{i=1}^{n_i} \log(\hat{p}(x_i)) + \frac{\lambda}{n_o} \sum_{j=1}^{n_o} \log(\hat{p}(z_j)) \right\}, \quad (6)$$

with

$$\hat{p}(y|x) = \frac{\hat{p}(y|x,i)\hat{p}(x|i) + \frac{\lambda}{M}\hat{p}(x|o)}{\hat{p}(x|i) + \lambda\hat{p}(x|o)} \quad \text{and} \quad \hat{p}(x) = \frac{1}{\lambda+1}\left(\hat{p}(x|i) + \lambda\hat{p}(x|o)\right). \quad (7)$$

Due to the bounds derived in Section 3, we denote our method by **Certified Certain Uncertainty (CCU)**. Note that if one uses a standard neural network model with softmax, i.e. $\hat{p}(y|x) = \hat{p}(y|x,i) = \frac{e^{f_y(x)}}{\sum_{m=1}^{M} e^{f_m(x)}}$, then the first term in equation 6 would be the cross-entropy loss for the in-distribution data and the second term the cross entropy loss for the out-distribution data with a uniform distribution over the classes. For this choice of $\hat{p}(y|x)$ and neglecting the terms for $\hat{p}(x)$ we recover the approach of Hein et al. (2019); Hendrycks et al. (2019) for training a classifier which outputs uniform confidence predictions on out-distribution data where $\frac{\hat{p}(i)}{\hat{p}(o)}$ corresponds to that regularization parameter $\lambda$. The key difference in our approach is that $\hat{p}(y|x) \neq \hat{p}(y|x, i)$ and the estimated densities for in- and out distribution $\hat{p}(x|i)$ and $\hat{p}(x|o)$ lead to a confidence calibration of $\hat{p}(y|x)$, and in turn the fit of the classifier influences the estimation of $\hat{p}(x|i)$ and $\hat{p}(x|o)$. The major advantage of our model is that we can give guarantees on the confidence of the classifier decision far away from the training data.

## 3 Provable guarantees for close to uniform predictions far away from the training data

In this section we provide two types of guarantees on the confidence of a classifier trained according to our model in equation 6. The first one says that the classifier has provably low confidence far away

from the training data, where an explicit bound on the minimal distance is provided, and the second provides an upper bound on the confidence in a ball around a given input point. The latter bound resembles robustness guarantees for adversarial samples (Hein & Andriushchenko, 2017; Wong & Kolter, 2018; Raghunathan et al., 2018; Mirman et al., 2018) and is quite different from the purely empirical evaluation done in OOD detection papers as we show in Section 4.

We provide our bounds for a more general mixture model which includes our GMM in equation 4 as a special case. To our knowledge, these are the first such bounds for neural networks and thus it is the first modification of a ReLU neural network so that it provably "knows when it does not know" (Hein et al., 2019) in the sense that far away from the training data the predictions are close to uniform over the classes.

**Theorem 3.1.** *Let $(x_i^{(i)}, y_i^{(i)})_{i=1}^n$ be the training set of the in-distribution and let the model for the conditional probability be given as*

$$\forall x \in \mathbb{R}^d, \ y \in \{1, \ldots, M\}, \qquad \hat{p}(y|x) = \frac{\hat{p}(y|x, i)\hat{p}(x|i) + \frac{\lambda}{M}\hat{p}(x|o)}{\hat{p}(x|i) + \lambda\hat{p}(x|o)}, \tag{8}$$

*where $\lambda = \frac{\hat{p}(o)}{\hat{p}(i)} > 0$ and let the model for the marginal density of the in-distribution $\hat{p}(x|i)$ and out-distribution $p(x|o)$ be given by the generalized GMMs*

$$\hat{p}(x|i) = \sum_{k=0}^{K_i} \alpha_k \exp\left(-\frac{d(x, \mu_k)^2}{2\sigma_k^2}\right), \qquad \hat{p}(x|o) = \sum_{l=0}^{K_o} \beta_l \exp\left(-\frac{d(x, \nu_l)^2}{2\theta_l^2}\right)$$

*with $\alpha_k, \beta_l > 0$ and $\mu_k, \nu_l \in \mathbb{R}^d \ \forall k = 1, \ldots K_i, \ l = 1, \ldots, K_o$ and $d : \mathbb{R}^d \times \mathbb{R}^d \to \mathbb{R}_+$ a metric. Let $z \in \mathbb{R}^d$ and define $k^* = \underset{k=1,\ldots,K_i}{\arg\min} \frac{d(z, \mu_k)}{\sigma_k}$, $i^* = \underset{i=1,\ldots,n}{\arg\min} d(z, x_i)$, $l^* = \underset{l=1,\ldots,K_o}{\arg\max} \beta_l \exp\left(-\frac{d(z, \nu_l)^2}{2\theta_l^2}\right)$ and $\Delta = \frac{\theta_{l^*}^2}{\sigma_{k^*}^2} - 1$. For any $\epsilon > 0$, if $\min_l \theta_l > \max_k \sigma_k$ and*

$$\min_{i=1,\ldots,n} d(z, x_i) \geq d(x_{i^*}, \mu_{k^*}) + d(\mu_{k^*}, \nu_{l^*})\left[\frac{2}{\Delta} + \frac{1}{\sqrt{\Delta}}\right] + \theta_{l^*}\sqrt{\frac{2}{\Delta}\log\left(\frac{M-1}{\epsilon\lambda}\frac{\sum_k \alpha_k}{\beta_{l^*}}\right)}, \tag{9}$$

*then it holds for all $m \in \{1, \ldots, M\}$ that*

$$\hat{p}(m|z) \leq \frac{1}{M}\left(1 + \epsilon\right). \tag{10}$$

*In particular, if $\min_i d(z, x_i) \to \infty$, then $\hat{p}(m|z) \to \frac{1}{M}$.*

The proof is given in Appendix A. Theorem 3.1 holds for any multi-class classifier which defines for each input a probability distribution over the labels. Given the parameters of the GMM's it quantifies at which distance of an input $z$ to the training set the classifier achieves close to uniform confidence. The theorem holds even if we use ReLU classifiers which in their unmodified form have been shown to produce arbitrarily high confidence far away from the training data Hein et al. (2019). This is a first step towards neural networks which provably know when they don't know.

In the next corollary, we provide an upper bound on the confidence over a ball around a given data point. This allows to give "confidence guarantees" for a whole volume and thus is much stronger than the usual pointwise evaluation of OOD methods.

**Corollary 3.1.** *Let $x_0 \in \mathbb{R}^d$ and $R > 0$, then with $\lambda = \frac{\hat{p}(o)}{\hat{p}(i)}$ it holds*

$$\max_{d(x, x_0) \leq R} \hat{p}(y|x) \leq \frac{1}{M}\frac{1 + M\frac{b}{\lambda}}{1 + \frac{b}{\lambda}}, \tag{11}$$

*where $b = \dfrac{\sum_{k=1}^{K_i} \alpha_k \exp\left(-\frac{\max\{d(x_0, \mu_k) - R, 0\}^2}{2\sigma_k^2}\right)}{\sum_{l=1}^{K_o} \beta_l \exp\left(-\frac{(d(x_0, \nu_l) + R)^2}{2\theta_l^2}\right)}$.*

The proof is in the Appendix B. We show in Section 4 that even though OOD methods achieve low confidence on noise images, the maximization of the confidence in a ball around a noise point (adversarial noise) yields high confidence predictions for OOD methods, whereas our classifier has provably low confidence, as certified by Corollary 3.1. The failure of OOD methods shows that the certification of entire regions is an important contribution of CCU which goes beyond the purely sampling-based evaluation.

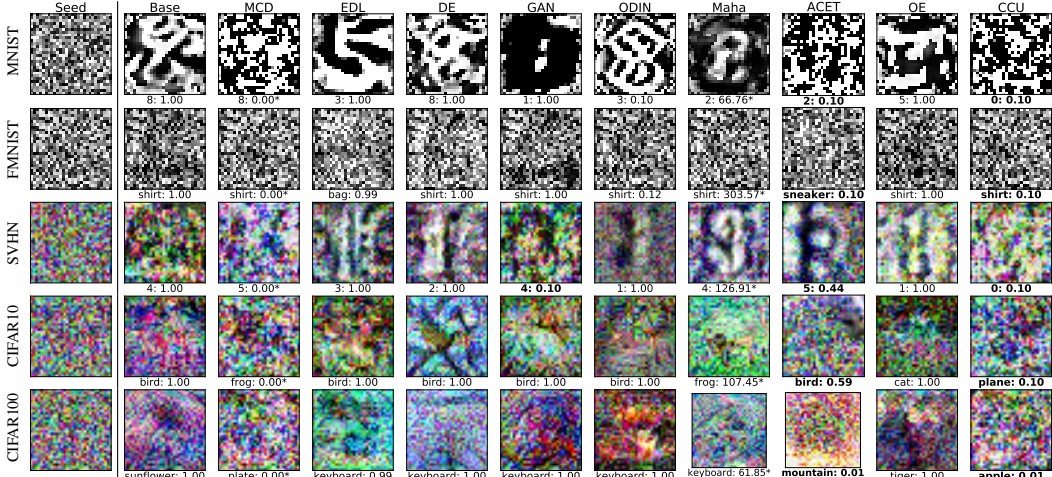

Figure 2: **Adversarial Noise:** We maximize the confidence of the OOD methods using PGD in the ball around a uniform noise sample (seed images, left) on which CCU is guaranteed by Corollary 3.1 to yield less than $1.1\frac{1}{M}$ maximal confidence. For each OOD method we report the image with the highest confidence. Maha and MCD use scores where lower is more confident (indicated by $*$). If we do *not* find a sample that has higher confidence/lower score than the median of the in-distribution, we highlight this in boldface. All other OOD methods fail on some dataset, see Table 1 for a quantitative version. ODIN at high temperatures always returns low confidence, so a value of 0.1 is not informative.

## 4 EXPERIMENTS

We evaluate the worst-case performance of various OOD detection methods within regions for which CCU yields guarantees and by standard OOD on MNIST (LeCun et al., 1998), FashionMNIST (Xiao et al., 2017), SVHN (Netzer et al., 2011), CIFAR10 and CIFAR100 (Krizhevsky & Hinton, 2009). We show that all other OOD methods yield undesired high confidence predictions in the certified low confidence regions of CCU and thus would not detect these inputs as out-distribution. For calibrating hyperparameters resp. training we use for all OOD methods the 80 Million Tiny Images (Torralba et al., 2008) as out-distribution Hendrycks et al. (2019) which yields a fair and realistic comparison.

**CCU:** As the Euclidean metric is known to be a relatively bad distance between two images we instead use the distance $d(x,y) = \left\| C^{-\frac{1}{2}}(x - y) \right\|$, where $C$ is generated as follows. We calculate the covariance matrix $C'$ on augmented in-distribution samples (see C.1). Let $(\lambda_i, u_i)_{i=1}^d$ be the eigenvalues/eigenvectors of $C'$. Then we set

$$C = \sum_{i=1}^{d} \max\{\lambda_i, 10^{-6} \max_j \lambda_j\} u_i u_i^T, \tag{12}$$

that is we fix a lower bound on the smallest eigenvalue so that $C$ has full rank. In Hendrycks & Gimpel (2017b) a similar metric has been used for detection of adversarial images. We choose $K_i = K_o = 100$ as the number of centroids for the GMMs. We initialize the in-GMM on augmented in-data using the EM algorithm with spherical covariance matrices in the transformed space, as in equation 4. For the out-distribution we use a subset of 20000 points for the initialization. While, initially it holds that $\forall k, l : \sigma_k < \theta_l$, as required in Theorem 3.1, this is not guaranteed during the optimization of equation 6. Thus, we enforce the constraint during training by setting: $\theta_l \mapsto \max\{\theta_l, 2 \max_k \sigma_k\}$ at every gradient step. Since the "classifier" and "density" terms in equation 6 have very different magnitudes we choose a small learning rate of $1e - 5$ for the parameters in the GMMs. It is also crucial to not apply weight decay to these parameters. The other hyperparameters are chosen as in the base model below.

**Benchmarks:** For all OOD methods we use LeNet on MNIST and a Resnet18 (for GAN and MCD we use VGG) otherwise. The hyperparameters used during training can be found in Appendix

| | | Base | MCD | EDL | DE | GAN | ODIN | Maha | ACET | OE | CCU |
|---|---|---|---|---|---|---|---|---|---|---|---|
| MNIST | TE | 0.5 | 0.4 | 0.4 | 0.4 | 0.8 | 0.5 | 0.9 | 0.6 | 0.7 | 0.6 |
| | SR | 100.0 | 99.0 | 100.0 | 100.0 | 43.5 | 100.0 | 100.0 | **0.0** | 100.0 | **0.0** |
| | AUC | 1.4 | 8.6 | 0.0 | 7.3 | 54.4 | 0.0 | 11.7 | **100.0** | 35.2 | **100.0** |
| FMNIST | TE | 4.8 | 5.8 | 5.2 | 4.9 | 5.7 | 4.8 | 4.8 | 4.8 | 5.7 | 4.9 |
| | SR | 100.0 | 72.5 | 100.0 | 100.0 | 99.0 | 100.0 | 100.0 | **0.0** | 100.0 | **0.0** |
| | AUC | 0.0 | 47.1 | 0.0 | 0.4 | 39.5 | 0.0 | 18.8 | **100.0** | 35.7 | **100.0** |
| SVHN | TE | 2.9 | 3.9 | 3.1 | 2.4 | 4.2 | 2.9 | 2.9 | 3.2 | 4.1 | 3.0 |
| | SR | 100.0 | 73.5 | 100.0 | 100.0 | **0.0** | 100.0 | 100.0 | 3.0 | 100.0 | **0.0** |
| | AUC | 0.0 | 34.1 | 0.0 | 0.0 | **100.0** | 0.0 | 0.0 | 96.5 | 0.0 | **100.0** |
| CIFAR10 | TE | 5.6 | 11.7 | 7.0 | 6.7 | 11.7 | 5.6 | 5.6 | 6.1 | 4.7 | 5.8 |
| | SR | 100.0 | 90.5 | 100.0 | 100.0 | 100.0 | 100.0 | 100.0 | 0.0 | 100.0 | **0.0** |
| | AUC | 0.0 | 23.9 | 0.0 | 0.0 | 25.3 | 0.0 | 0.0 | 99.9 | 0.0 | **100.0** |
| CIFAR100 | TE | 23.3 | 45.3 | 31.1 | 27.5 | 43.8 | 23.3 | 23.2 | 25.2 | 24.7 | 25.9 |
| | SR | 100.0 | 100.0 | 100.0 | 100.0 | 89.5 | 100.0 | 100.0 | 3.5 | 100.0 | **0.0** |
| | AUC | 0.1 | 17.3 | 0.0 | 0.2 | 15.3 | 0.0 | 0.0 | 95.8 | 2.5 | **100.0** |

Table 1: Worst-case performance of different OOD methods in neighborhoods around uniform noise points certified by CCU. We report the clean test error (TE) on the in-distribution (GAN and MCD use VGG). The success rate (SR) is the fraction of adversarial noise points for which the confidence/score inside the ball is higher than the median of the in-distribution's confidence/score. The AUC quantifies detection of adversarial noise versus in-distribution. All values in %.

C. The AUC (area under ROC) is computed by treating in-distribution versus out-distribution as a two-class problem using the confidence/score of the method as criterion. Alternatively one could report the AUPR (area under precision-recall curve) which we do in Appendix G. **MCD:** Monte-Carlo Dropout (Gal & Ghahramani, 2016) uses dropout at train and at test time. Since it is not clear where to put the dropout layers in a ResNet, we use VGG instead. We take the softmax from 7 forward passes (Shafaei et al., 2018) and use the mean of the output for prediction and the variance as score. **EDL:** Evidential deep learning (Sensoy et al., 2018) replaces the softmax layer of a neural network and introduces a different loss function that encourages better uncertainty estimates. **DE:** Deep ensembles (Lakshminarayanan et al., 2017b) average the softmax outputs of five models that were adversarially trained via FGSM (Goodfellow et al., 2015) with step size $\epsilon = 0.01$. **GAN:** The framework of confidence-calibrated classifiers (Lee et al., 2017) relies on training a GAN alongside a classifier such that the GAN's generator is encouraged to generate points close to but not on the in-distribution. On these points one then enforces uniform confidence. We used their provided code to train a VGG this way, as we were unable to adapt the method to a ResNet with an acceptable test error (e.g. TE$< 30\%$ on SVHN). **ODIN:** ODIN (Liang et al., 2017) consists of two parts: a temperature $T$ by which one rescales the logits before the softmax layer $\frac{e^{f_n/T}}{\sum_k e^{f_k/T}}$ and a preprocessing step that applies a single FGSM-step (Goodfellow et al., 2015) of length $\epsilon$ before evaluating the input. The two parameters are calibrated on the out-distribution. **Maha:** The approach in Lee et al. (2018c) is based on computing a class-conditional Mahalanobis distance in feature space and applying an ODIN-like preprocessing step for each layer. Following Ren et al. (2019) we use a single-layer version of Lee et al. (2018c) on our networks' penultimate layers because the multi-layer version in the original code does not support gradient-based attacks. **OE:** Outlier exposure (Hendrycks et al., 2019) enforces uniform confidence on a large out-distribution. We use their provided code to train a model with our chosen architecture. **ACET:** Adversarial confidence enhanced training (ACET) (Hein et al., 2019) enforces low confidence on a ball around points from an out-distribution by running adversarial attacks during training. In order to make the comparison with OE more meaningful we use 80M tiny images to draw the seeds rather than smoothed uniform noise as in Hein et al. (2019). We refer to Appendix F for a discussion of the influence of this choice on the results.

Some of the above OOD papers optimize their hyperparameters on a validation set for each out-distribution they test on. However, this leads to different classifiers for each out-distribution dataset

| | | Base | MCD | EDL | DE | GAN | ODIN | Maha | ACET | OE | CCU |
|---|---|---|---|---|---|---|---|---|---|---|---|
| MNIST | FMNIST | 97.4 | 93.1 | 99.3 | 99.2 | 99.4 | 98.7 | 96.8 | **100.0** | 99.9 | 99.9 |
| | EMNIST | 89.2 | 82.0 | 89.0 | 92.1 | 92.8 | 88.9 | 91.6 | 95.0 | **95.8** | 92.0 |
| | GrCIFAR10 | 99.7 | 94.7 | 99.7 | **100.0** | 99.1 | 99.9 | 98.7 | **100.0** | **100.0** | **100.0** |
| | Noise | **100.0** | 95.2 | 99.9 | **100.0** | 99.3 | **100.0** | 97.2 | **100.0** | **100.0** | **100.0** |
| | Uniform | 95.2 | 87.9 | 99.9 | 97.9 | 99.9 | 98.2 | **100.0** | **100.0** | **100.0** | **100.0** |
| FMNIST | MNIST | 96.7 | 82.7 | 94.5 | 96.7 | **99.9** | 99.0 | 96.7 | 96.4 | 96.3 | 97.8 |
| | EMNIST | 97.5 | 87.3 | 95.6 | 97.1 | **99.9** | 99.3 | 97.5 | 97.6 | 99.3 | 99.5 |
| | GrCIFAR10 | 91.0 | 92.3 | 84.0 | 86.1 | 85.3 | 93.0 | 98.2 | 96.2 | **100.0** | **100.0** |
| | Noise | 97.3 | 94.0 | 95.6 | 97.4 | 98.9 | 98.9 | 98.9 | 97.8 | **100.0** | **100.0** |
| | Uniform | 96.9 | 93.3 | 95.6 | 98.3 | 93.2 | 98.8 | 99.1 | **100.0** | 97.6 | **100.0** |
| SVHN | CIFAR10 | 95.4 | 91.9 | 95.9 | 97.9 | 96.8 | 95.9 | 97.1 | 95.2 | **100.0** | **100.0** |
| | CIFAR100 | 94.5 | 91.4 | 95.6 | 97.6 | 96.1 | 94.8 | 96.7 | 94.8 | **100.0** | **100.0** |
| | LSUN_CR | 95.6 | 92.0 | 95.3 | 97.9 | 99.0 | 96.5 | 97.2 | 97.1 | **100.0** | **100.0** |
| | Imagenet- | 94.7 | 91.8 | 95.7 | 97.7 | 97.8 | 95.1 | 96.8 | 97.3 | **100.0** | **100.0** |
| | Noise | 96.4 | 93.1 | 97.1 | **98.2** | 96.2 | 82.7 | 98.0 | 95.8 | 97.8 | 97.4 |
| | Uniform | 96.8 | 93.1 | 96.5 | 95.6 | **100.0** | 97.9 | 97.8 | **100.0** | **100.0** | **100.0** |
| CIFAR10 | SVHN | 95.8 | 81.9 | 92.3 | 90.3 | 83.9 | 96.7 | 91.5 | 93.7 | **98.8** | 98.2 |
| | CIFAR100 | 87.3 | 78.6 | 87.3 | 88.2 | 82.9 | 87.5 | 82.8 | 86.9 | **95.3** | 94.2 |
| | LSUN_CR | 91.9 | 81.3 | 90.8 | 92.0 | 89.9 | 93.3 | 89.2 | 91.2 | **98.6** | 98.2 |
| | Imagenet- | 87.5 | 78.4 | 88.2 | 87.7 | 84.0 | 88.1 | 84.1 | 86.5 | **94.7** | 93.3 |
| | Noise | 96.5 | 79.9 | 88.9 | 90.3 | 81.8 | **97.6** | 94.4 | 94.8 | 97.3 | 97.0 |
| | Uniform | 96.8 | 81.0 | 89.9 | 96.6 | 73.0 | 98.8 | **100.0** | **100.0** | 98.8 | **100.0** |
| CIFAR100 | SVHN | 78.8 | 59.2 | 80.4 | 83.2 | 75.9 | 81.3 | 77.5 | 73.9 | 93.5 | **94.2** |
| | CIFAR10 | 78.6 | 58.9 | 73.3 | 76.3 | 69.3 | 79.5 | 59.9 | 77.2 | **81.6** | 80.2 |
| | LSUN_CR | 81.0 | 59.4 | 74.2 | 81.6 | 79.8 | 81.4 | 79.7 | 78.0 | 95.4 | **95.9** |
| | Imagenet- | 80.8 | 59.2 | 76.0 | 78.2 | 73.9 | 81.3 | 70.8 | 79.5 | **83.8** | 81.4 |
| | Noise | 73.4 | 58.7 | 65.9 | 67.5 | 73.6 | 76.8 | 90.6 | 62.9 | 86.9 | **94.6** |
| | Uniform | 93.3 | 62.0 | 29.8 | 36.6 | **100.0** | 93.5 | 94.3 | **100.0** | 99.1 | **100.0** |

Table 2: AUC (in- versus out-distribution detection based on confidence/score) in percent for different OOD methods and datasets (higher is better). OE and CCU have the best OOD performance.

which seems unrealistic as we want to have good generic OOD performance and not for a particular dataset. Thus we keep the comparison realistic and fair by calibrating the hyperparameters of all methods on a subset of 80M tiny images and then evaluating on the other unseen distributions.

**Certified robustness against adversarial noise:** We sample uniform noise images as they are obviously out-distribution for all tasks and certify using Corollary 3.1 the largest ball around the uniform noise sample on which CCU attains at most $1.1\cdot$ uniform confidence, that is $1.1\%$ on CIFAR100 and $11\%$ on all other datasets. We describe how to compute the radius of this ball in Appendix D. In principle it could be possible that the certified balls contain training or test images. In Appendix E we show that this is not the case. We construct adversarial noise samples for all OOD methods by maximizing the confidence/minimizing the score via a PGD attack with 500 steps and 50 random restarts on this ball. Further details of the attack can be found in Appendix C.2. In Table 1 we show the results of running this attack on the different models. We used 200 noise images and we report clean test error on the in-distribution, the success rate (SR) (fraction of adversarial noise points for which the confidence resp. score inside the ball is higher resp. lower than the median of the in-distribution's confidence/score) and the AUC for the separation of the generated adversarial noise images and the in-distribution based on confidence/score. By construction, see Corollary 3.1, our method provably makes no overconfident predictions but we nevertheless run the attack on CCU as well. We note that only CCU performs perfectly on this task for all datasets - all other OOD methods fail at least on one dataset, most of them on all. We also see that ACET achieves very robust performance which may be expected as it does some kind of adversarial training for OOD detection. Nevertheless, even though they are very rare, high-confidence adversarial noise images for ACET can be found on SVHN, CIFAR10 and CIFAR100 and ACET has no guarantees. We illustrate the generated adversarial noise images for all methods in Figure 2.

**OOD performance:** For each dataset and method we report the AUC for the binary classification problem of discriminating in- and out-distribution based on confidence resp. score. The results are shown in Table 2. The list of datasets we use for OOD detection can be seen in Table 2. LSUN_CR refers to only the classroom class of LSUN and Imagenet- is a subset of 10000 resized Imagenet validation images, that have no overlap with CIFAR10/CIFAR100 classes. The noise dataset was obtained as in Hein et al. (2019) by first shuffling the pixels of the test images in the in-distribution and then smoothing them by a Gaussian filter of uniformly random width, followed by a rescaling so that the images have full range. GrCIFAR10 refers to the images in CIFAR10 being grayscaled and resized to 28x28 and Uniform describes images sampled uniformly at random from $[0, 1]^d$. We see that OE and CCU have the best OOD performance. MCD is worse than the base model which confirms the results found in Leibig et al. (2017) that MCD is not useful for OOD. DE outperforms EDL but is not much better than the baseline for CIFAR10 and CIFAR100. The performance of Maha is worse than what has been reported in Lee et al. (2018c) which can have two reasons. We just use their version where one uses the scores only from the last layer and we do not calibrate hyperparameters for each test set separately but just once on the Tiny Image dataset. Especially on CIFAR10 we found that the results depend strongly on the step size. The results of ACET, GAN and ODIN are mixed but clearly outperform the baseline. Comparing Table 1 and Table 2 we see that most models perform well when evaluating on uniform noise but fail when finding the worst case in a small neighborhood around the noise point. Thus we think that such worst-case analysis should become standard in OOD evaluation.

## 5 CONCLUSION

In Hein et al. (2019) it has recently been shown that ReLU networks produce arbitrarily highly confident predictions far away from the training data, which could only be resolved by a modification of the network architecture. With CCU we present such a modification which explicitly integrates a generative model and provably show that the resulting neural network produces close to uniform predictions far away from the training data. Moreover, CCU is the only OOD method which can guarantee low confidence predictions over a whole volume rather than just pointwise and we show that all other OOD methods fail in this worst-case setting. CCU achieves this without loss in test accuracy or OOD performance. In the future it would be interesting to use more powerful generative models for which one can also guarantee their behavior far away from the training data. This is currently not the case for VAEs and GANs (Nalisnick et al., 2019; Hendrycks et al., 2019).

### ACKNOWLEDGMENTS

The author acknowledge support from the BMBF through the Tübingen AI Center (FKZ: 01IS18039A) and by the DFG TRR 248, project number 389792660 and the DFG Excellence Cluster "Machine Learning -New Perspectives for Science", EXC 2064/1, project number 390727645. The authors thank the International Max Planck Research School for Intelligent Systems (IMPRS-IS) for supporting Alexander Meinke.

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

## A  APPENDIX - PROOF OF THEOREM 3.1

**Theorem 3.1.** *Let $(x_i, y_i)_{i=1}^n$ be the training set of the training distribution. We define the model for the conditional probability over the classes $y \in \{1, \ldots, M\}$ given $x$ as*

$$\hat{p}(y|x) = \frac{\hat{p}(y|x,i)\hat{p}(x|i) + \frac{\lambda}{M}\hat{p}(x|o)}{\hat{p}(x|i) + \lambda\hat{p}(x|o)}, \tag{13}$$

*where $\lambda = \frac{\hat{p}(o)}{\hat{p}(i)} > 0$ and $M > 1$. Further, let the model for the marginal density of the in-distribution $\hat{p}(x|i)$ and out-distribution $p(x|o)$ be given by the generalized GMMs*

$$\hat{p}(x|i) = \sum_{k=0}^{K_i} \alpha_k \exp\left(-\frac{d(x,\mu_k)^2}{2\sigma_k^2}\right), \qquad \hat{p}(x|o) = \sum_{l=0}^{K_o} \beta_l \exp\left(-\frac{d(x,\nu_l)^2}{2\theta_l^2}\right)$$

*with $\alpha_k, \beta_l > 0$ and $\mu_k, \nu_l \in \mathbb{R}^d \quad \forall k = 1, \ldots K_i, \ l = 1, \ldots, K_o$ and $d : \mathbb{R}^d \times \mathbb{R}^d \to \mathbb{R}_+$ a metric. Let $z \in \mathbb{R}^d$ and define $k^* = \underset{k=1,\ldots,K_i}{\arg\min} \frac{d(z,\mu_k)}{\sigma_k}$, $i^* = \underset{i=1,\ldots,n}{\arg\min} d(z,x_i)$, $l^* = \underset{l=1,\ldots,K_o}{\arg\min} \beta_l \exp\left(-\frac{d(z,\nu_l)^2}{2\theta_l^2}\right)$ and $\Delta = \frac{\theta_{l^*}^2}{\sigma_{k^*}^2} - 1$. For any $\epsilon > 0$, if $\min_l \theta_l > \max_k \sigma_k$ and*

$$\min_{i=1,\ldots,n} d(z,x_i) \geq d(x_{i^*},\mu_{k^*}) + d(\mu_{k^*},\nu_{l^*})\left[\frac{2}{\Delta} + \frac{1}{\sqrt{\Delta}}\right] + \theta_{l^*}\sqrt{\frac{2}{\Delta}\log\left(\frac{M-1}{\epsilon\,\lambda}\frac{\sum_k \alpha_k}{\beta_{l^*}}\right)}, \tag{14}$$

*then it holds for all $m \in \{1, \ldots, M\}$ that*

$$\hat{p}(m|z) \leq \frac{1}{M}\big(1 + \epsilon\big). \tag{15}$$

*In particular, if $\min_i d(z,x_i) \to \infty$, then $\hat{p}(m|z) \to \frac{1}{M}$.*

*Proof.* The proof essentially hinges on upper bounding $\frac{\hat{p}(z|i)}{\hat{p}(z|o)}$ using the specific properties of the Gaussian mixture model. We note that

$$\hat{p}(y|x) = \frac{\hat{p}(y|x,i)\hat{p}(x|i) + \frac{\lambda}{M}\hat{p}(x|o)}{\hat{p}(x|i) + \lambda\hat{p}(x|o)} = \frac{1}{M}\frac{1 + \frac{M}{\lambda}\frac{\hat{p}(x|i)}{\hat{p}(x|o)}}{1 + \frac{1}{\lambda}\frac{\hat{p}(x|i)}{\hat{p}(x|o)}} \leq \frac{1}{M}\left(1 + \frac{M-1}{\lambda}\frac{\hat{p}(x|i)}{\hat{p}(x|o)}\right)$$

The last step holds because the function $g(\xi) = \frac{1+M\xi}{1+\xi}$ is monotonically increasing

$$\frac{\partial g}{\partial \xi} = \frac{M-1}{(1+\xi)^2} \quad \text{and} \quad \frac{\partial^2 g}{\partial \xi^2} = -2\frac{M-1}{(1+\xi)^3}. \tag{16}$$

As the second deriviative is negative for $\xi \geq 0$, $g$ is concave for $\xi \geq 0$ and thus

$$\frac{1+M\xi}{1+\xi} = g(\xi) \leq g(0) + \frac{\partial g}{\partial \xi}\bigg|_{\xi=0}(\xi - 0) = 1 + (M-1)\xi. \tag{17}$$

In order to achieve the required result we need to show that $\frac{M-1}{\lambda}\frac{\hat{p}(x|i)}{\hat{p}(x|o)} \leq \epsilon$ for $x$ sufficiently far away from the training data.

We note that

$$\frac{\hat{p}(x|i)}{\hat{p}(x|o)} = \frac{\sum_k \alpha_k \exp\left(-\frac{d(x,\mu_k)^2}{2\sigma_k^2}\right)}{\sum_l \beta_l \exp\left(-\frac{d(x,\nu_l)^2}{2\theta_l^2}\right)} \leq \frac{\max_k \exp\left(-\frac{d(x,\mu_k)^2}{2\sigma_k^2}\right)\sum_k \alpha_k}{\max_l \beta_l \exp\left(-\frac{d(x,\nu_l)^2}{2\theta_l^2}\right)}$$

$$= \frac{\sum_k \alpha_k}{\beta_{l^*}}\exp\left(-\frac{d(x,\mu_{k^*})^2}{2\sigma_{k^*}^2} + \frac{d(x,\nu_{l^*})^2}{2\theta_{l^*}^2}\right)$$

where $k^* = \arg\min_k \frac{d(x,\mu_k)^2}{2\sigma_k^2}$ and $l^* = \arg\max_l \beta_l \exp(-\frac{d(x,\nu_l)^2}{2\theta_l^2})$. Using the triangle inequality, $d(x,\nu_{l*}) \leq d(x,\mu_{k*}) + d(\mu_{k*},\nu_{l*})$, we get the desired condition as

$$\frac{\sum_k \alpha_k}{\beta_{l*}} \exp\left(-d(x,\mu_{k*})^2\left(\frac{1}{2\sigma_{k*}^2} - \frac{1}{2\theta_{l*}^2}\right) + \frac{d(\mu_{k*},\nu_{l*})d(x,\mu_{k*})}{\theta_{l*}^2} + \frac{d(\mu_{k*},\nu_{l*})^2}{2\theta_{l*}^2}\right) \leq \frac{\epsilon\,\lambda}{M-1}$$

Thus we get with $a = \left(\frac{1}{2\sigma_{k*}^2} - \frac{1}{2\theta_{l*}^2}\right)$, $b = \frac{d(\mu_{k*},\nu_{l*})}{\theta_{l*}^2}$ and $c = \frac{d(\mu_{k*},\nu_{l*})^2}{2\theta_{l*}^2}$, $d = \log\left(\frac{\epsilon\lambda}{M-1}\frac{\beta_{l*}}{\sum_k \alpha_k}\right)$, the quadratic inequality

$$-d(x,\mu_{k*})^2 a + d(x,\mu_{k*})b + c \leq d,$$

where $d < 0$ for sufficiently small $\epsilon$. We get the solution

$$d(x,\mu_{k*}) \geq \frac{b}{2a} + \sqrt{\max\left\{0, \frac{c-d}{a} + \frac{b^2}{4a^2}\right\}}.$$

It holds, using $\sqrt{a+b} \leq \sqrt{a} + \sqrt{b}$ for $a,b > 0$,

$$\frac{b}{2a} + \sqrt{\max\left\{0, \frac{c-d}{a} + \frac{b^2}{4a^2}\right\}} \leq \frac{b}{a} + \sqrt{\frac{c}{a}} + \sqrt{\frac{-d}{a}}.$$

One can simplify

$$\frac{b}{a} = 2\frac{\sigma_{k*}^2\theta_{l*}^2}{\theta_{l*}^2 - \sigma_{k*}^2}\frac{d(\mu_{k*},\nu_{l*})}{\theta_{l*}^2} = 2\frac{\sigma_{k*}^2 d(\mu_{k*},\nu_{l*})}{\theta_{l*}^2 - \sigma_{k*}^2} = 2\frac{d(\mu_{k*},\nu_{l*})}{\frac{\theta_{l*}^2}{\sigma_{k*}^2} - 1}$$

$$\frac{c}{a} = 2\frac{\sigma_{k*}^2\theta_{l*}^2}{\theta_{l*}^2 - \sigma_{k*}^2}\frac{d(\mu_{k*},\nu_{l*})^2}{2\theta_{l*}^2} = \frac{\sigma_{k*}^2 d(\mu_{k*},\nu_{l*})^2}{\theta_{l*}^2 - \sigma_{k*}^2} = \frac{d(\mu_{k*},\nu_{l*})^2}{\frac{\theta_{l*}^2}{\sigma_{k*}^2} - 1}$$

Noting that $d(x,\mu_{k*}) \geq |d(x,x_{i*}) - d(x_{i*},\mu_{k*})|$ we get that

$$d(x,x_{i*}) \geq d(x_{i*},\mu_{k*}) + \frac{b}{a} + \sqrt{\frac{c}{a}} + \sqrt{\frac{-d}{a}},$$

implies $\frac{M-1}{\lambda}\frac{\hat{p}(x|i)}{\hat{p}(x|o)} \leq \epsilon$. The last statement follows directly by noting that by assumption $a > 0$ (independently of the choice of $l^*$ and $k^*$) and $b,c,d(x_{i*},\mu_{k*})$ are bounded as $K_i, K_o, n$ are finite. With $\Delta = \frac{\theta_{l*}^2}{\sigma_{k*}^2} - 1$ we can rewrite the required condition as

$$d(x,x_{i*}) \geq d(x_{i*},\mu_{k*}) + d(\mu_{k*},\nu_{l*})\left[\frac{2}{\Delta} + \frac{1}{\sqrt{\Delta}}\right] + \theta_{l*}\sqrt{\frac{2}{\Delta}\log\left(\frac{M-1}{\epsilon\,\lambda}\frac{\sum_k \alpha_k}{\beta_{l*}}\right)}.$$

$\square$

## B APPENDIX - PROOF OF COROLLARY 3.1

**Corollary 3.1.** *Let* $x_0 \in \mathbb{R}^d$ *and* $R > 0$, *then with* $\lambda = \frac{\hat{p}(o)}{\hat{p}(i)}$ *it holds*

$$\max_{d(x,x_0)\leq R} \hat{p}(y|x) \leq \frac{1}{M}\frac{1 + M\frac{b}{\lambda}}{1 + \frac{b}{\lambda}}, \tag{18}$$

*where* $b = \dfrac{\sum_{k=1}^{K_i}\alpha_k\exp\left(-\frac{\max\{d(x_0,\mu_k)-R,0\}^2}{2\sigma_k^2}\right)}{\sum_{l=1}^{K_o}\beta_l\exp\left(-\frac{(d(x_0,\nu_l)+R)^2}{2\theta_l^2}\right)}$.

*Proof.* From the previous section we already know that $\hat{p}(y|x) \leq \frac{1}{M}\frac{1 + M\frac{b}{\lambda}}{1 + \frac{b}{\lambda}}$ as long as $\frac{p(x|i)}{p(x|o)} \leq b$. Now we can separately bound the numerator and denominator within a ball of radius $R$ around $x_0$.

For the numerator we have

$$\max_{d(x,x_0)\leq R} \hat{p}(x|i) \leq \sum_{k=1}^{K} \alpha_k \max_{d(x,x_0)\leq R} e^{-\frac{d(x,\mu_k)^2}{2\sigma_k^2}} \tag{19}$$

$$\leq \sum_{k=1}^{K} \alpha_k \exp\left(-\frac{\min_{d(x,x_0)\leq R} d(x,\mu_k)^2}{2\sigma_k^2}\right)$$

$$\leq \sum_{k=1}^{K} \alpha_k \exp\left(-\frac{(\max\{d(\mu_k,x_0)-R,0\})^2}{2\sigma_k^2}\right), \tag{20}$$

where we have lower bounded $\min_{d(x,x_0)\leq R} d(x,\mu_k)$ via the reverse triangle inequality

$$\min_{d(x,x_0)\leq R} d(x,\mu_k) \geq \min_{d(x,x_0)\leq R} |d(x_0,\mu_k)-d(x,x_0)|,$$

$$\geq \max\left\{\min_{d(x,x_0)\leq R}(d(x_0,\mu_k)-d(x_0,\mu_k)),0\right\},$$

$$\geq \max\{d(x_0,\mu_k)-r,0\}. \tag{21}$$

The denominator can similarly be bounded via

$$\min_{d(x,x_0)\leq R} \hat{p}(x|o) \geq \sum_{l=1}^{K_o} \beta_l \min_{d(x,x_0)\leq R} e^{-\frac{d(x,\nu_l)^2}{2\theta_k^2}} \tag{22}$$

$$\geq \sum_{l=1}^{K_o} \beta_l \exp\left(-\frac{\max_{d(x,x_0)\leq R} d(x,\nu_l)^2}{2\theta_l^2}\right)$$

$$\geq \sum_{l=1}^{K_o} \beta_l \exp\left(-\frac{(d(x_0,\nu_l)+R)^2}{2\theta_l^2}\right). \tag{23}$$

With both of these bounds in place the conclusion immediately follows. $\square$

## C  Appendix - Experimental Details

Unless specified otherwise we use ADAM on MNIST with a learning rate of $1e-3$ and SGD with learning rate $0.1$ for the other datasets. The learning rate for the GMM is always set to $1e-5$. We decrease all learning rates by a factor of 10 after 50, 75 and 90 epochs. Our batch size is 128, the total number of epochs 100 and weight decay is set to $5e-4$.

When training ACET, OE and CCU with 80 million tiny images we pick equal batches of in- and out-distribution data (corresponding to $p(i)=p(o)$) and concatenate them into a batches of size 256. Note that during the 100 epochs only a fraction of the 80 million tiny images are seen and so there is no risk of over-fitting.

### C.1  Data Augmentation

Our data augmentation scheme uses random crops with a padding of 2 pixels on MNIST and FMNIST. On SVHN, CIFAR10 and CIFAR100 the padding width is 4 pixels. For SVHN we fill the padding with the value at the boundary and for CIFAR we apply reflection at the boundary pixels. On top of this we include random horizontal flips on CIFAR. For MNIST and FMNIST we generate 60000 such samples and for SVHN and CIFAR 50000 samples by drawing from the clean dataset without replacement. This augmented data is used to calculate the covariance matrix from equation 12. During the actual training we use the same data augmentation scheme in a standard fashion.

### C.2  Attack details

We begin with a step size of 3 and for each of the 50 restarts we randomly initialize at some point in the ellipsoid. Whenever a gradient step successfully decreases the losses we increase the step size by

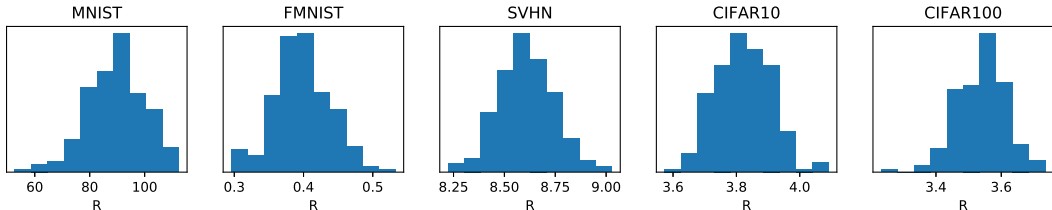

Figure 3: **Histograms of bounds:** Certified radius in transformed space for different datasets.

a factor of 1.1. Whenever the loss increases instead we use backtracking and decrease the step size by a factor of 2. We apply normal PGD using the $l_2$-norm in the transformed space to ensure that we stay on the ellipsoid and after each gradient step we rotate back into the original space to project onto the box $[0, 1]^d$. The result is not guaranteed be on the ellipsoid so after the 500 steps we use the alternating projection algorithm (Bauschke & Borwein, 1996) for 10 steps which is guaranteed to converge to a point in the intersection of the ellipsoid and the box because both of these sets are convex.

## D   APPENDIX - FINDING A THE CERTIFIABLE RADIUS

Since Corollary 3.1 does not explicitly give a radius, one has to numerically invert the bound. The bound

$$b(R) = \frac{\sum_{k=1}^{K_i} \alpha_k \exp\left(-\frac{\max\{d(x_0, \mu_k) - R, 0\}^2}{2\sigma_k^2}\right)}{\sum_{l=1}^{K_o} \beta_l \exp\left(-\frac{(d(x_0, \nu_l) + R)^2}{2\theta_l^2}\right)} \tag{24}$$

is monotonically increasing in $R$. Thus, for a given sample $x_0$ one can fix a desired bound $\max_{d(x,x_0) \leq R} \hat{p}(x|i) \leq \frac{1}{M}\nu$, where $\nu \in (1, M)$ and then find the unique solution

$$b(R) = \frac{\nu - 1}{M - \nu}\lambda \tag{25}$$

for $R$ via bisection. This radius $\hat{R}$ will then represent the maximal radius, that one can certify using Corollary 3.1. The presumption is, of course, that for $R = 0$ one has a sufficiently low bound in the first place, i.e. that a solution exists. In our experiments on uniform noise we did not encounter a single counterexample to this assumption. We show the radii for the different datasets in Figure 3.

## E   APPENDIX - ANALYSIS OF THE CERTIFIED BALLS AROUND UNIFORM NOISE IMAGES

As one can observe in Figure 2 the images which maximize the confidence in the certified ball around the uniform noise image are sometimes quite far away from the original noise image. As CCU certifies low confidence (the maximal confidence is less than $1.1 \times \frac{1}{M}$ - so the predicted probability distribution over the classes is very close to the uniform distribution) over the whole ball, it is a natural question what these balls look like and what kind of images they contain. In particular, it is in general not desired that the certified balls contain images from the training and test set. For each dataset we certified balls around 200 uniform noise images and for each of the certified balls we check if it contains training or test images of the corresponding dataset. We found that even though the certified balls are large, not a single training or test image was contained in any of them. This justifies the use of our proposed threat model.

A different problem could be that our threshold of $\frac{1.1}{M}$ for the certification is too high and that many predictions on the test set have confidence less than this threshold. For this purpose we report in Table 3 the smallest predicted confidence of CCU on the test set $T$, that is

$$\min_{x \in T} \max_{y \in \{1,...,M\}} \hat{p}(y|x),$$

|        | $\min p(y|x)$ | $\# < \frac{1.1}{M}$ | $\% < \frac{1.1}{M}$ |
|--------|---------------|----------------------|----------------------|
| MNIST    | 33.08 | 0   | 0    |
| FMNIST   | 28.77 | 0   | 0    |
| SVHN     | 10.02 | 20  | 0.08 |
| CIFAR10  | 10.01 | 529 | 5.29 |
| CIFAR100 | 1.03  | 130 | 1.30 |

Table 3: Lowest confidence that CCU attains on the test set (in percent) as well as total number of test points on which confidence is lower than our imposed bound of $\frac{1.1}{M}$.

for each dataset and the total number of test samples where the confidence is below $\frac{1.1}{M}$. While for MNIST and FMNIST, this never happens, and for SVHN this is negligible (less than $0.1\%$ of the test set), for CIFAR10 and CIFAR100 this happens in $5.3\%$ resp. $1.3\%$ of the cases.

In theory, this could impair our AUC value for the detection of adversarial noise. However, in practice our bound for the confidence is quite conservative as the bound is only tight in very specific configurations of the centroids of the Gaussian mixture model which are unlikely to happen for any practical dataset, meaning that the actual maximal confidence in the certified region is typically significantly lower. In fact the AUC values of CCU are always $100\%$ which means that for all 200 certified balls the maximal value of the confidence of CCU in any of these balls (found by our PGD attack algorithm) is lower than the minimal confidence of all predictions on the test set as reported in Table 3. On the other hand assuming here also a worst case scenario in the sense we assume that the upper bound of the maximal confidence is attained in all 200 certified balls, then the (certified) AUC value would be: 99.92% for SVHN, 94.71% for CIFAR10, and 98.70% for CIFAR100. Note that this theoretical lower bound on our performance is still better than all other models' empirical performance on this task, as reported in Table 1 on both CIFAR10 and CIFAR100, and only marginally below the perfect AUC of GAN on SVHN.

## F    Appendix - Performance of ACET when trained on adversarial uniform noise

Similar to our CCU, ACET Hein et al. (2019) requires that one chooses a model for the out-distribution in order to generate their "adversarial noise" during training. We trained the ACET model with the same out-distribution model as for all other models namely using the tiny image dataset as suggested in Hendrycks et al. (2019) with a PGD attack that starts at the original point and takes 40 FGSM steps in order to maximize the maximal confidence over the classes. We use backtracking and halve the step size whenever the loss does not increase. However, the authors of Hein et al. (2019) used smoothed uniform noise and a $l_\infty$-threat model during training. Since our worst case analysis for OOD is based on attacking uniform noise images, this suggests that training ACET with uniform noise should improve the performance of ACET for the worst case analysis. We report below the results of ACET2 (the original model in the paper using tiny images for training is called ACET) based on attacking using uniform noise images during training with a $l_\infty$-threat model with $\epsilon = 0.3$ as suggested in Hein et al. (2019). We report the normal OOD performance in Table 4 and the worst case analysis of adversarial noise in Table 5. While it is not surprising that ACET outperforms ACET2 on the standard OOD detection task in Table 4 as it has seen more realistic "noise" images during training, the worse performance of ACET2 for the worst case analysis in Table 5 is at first sight counter-intuitive. However, note that the threat model of the attacks in our worst case analysis is the Mahalanobis-type $l_2$-type metric, see 12, while ACET2 uses an $l_\infty$-attack model with $\epsilon = 0.3$ during training. As the size of the balls for the Mahalanobis-type $l_2$-type metric is quite large, there is not much overlap between the two sets. This explains why ACET2 fails here. In summary, we have shown that by using tiny images as out-distribution during training, ACET improves in terms of OOD detection performance over ACET2, which is similar to the version suggested in Hein et al. (2019).

## G    Appendix - Precision and Recall

In addition to the AUC presented in Table 2 we follow Hendrycks & Gimpel (2017c) and report the area under the precision/recall curve (AUPR). Precision at a specific threshold is defined as the

| MNIST | ACET | ACET2 |
|---|---|---|
| FMNIST | **100.0** | 99.8 |
| EMNIST | **95.0** | 93.5 |
| GrCIFAR10 | **100.0** | **100.0** |
| Noise | **100.0** | **100.0** |
| Uniform | **100.0** | **100.0** |

| FMNIST | ACET | ACET2 |
|---|---|---|
| MNIST | 96.4 | **96.5** |
| EMNIST | **97.6** | 97.3 |
| GrCIFAR10 | **96.2** | 91.6 |
| Noise | **97.8** | 97.1 |
| Uniform | **100.0** | **100.0** |

| SVHN | ACET | ACET2 |
|---|---|---|
| CIFAR10 | **95.2** | 94.2 |
| CIFAR100 | **94.8** | 93.7 |
| LSUN_CR | **97.1** | 96.1 |
| Imagenet- | **97.3** | 95.6 |
| Noise | **95.2** | **95.2** |
| Uniform | **100.0** | **100.0** |

| CIFAR10 | ACET | ACET2 |
|---|---|---|
| SVHN | **93.7** | 82.8 |
| CIFAR100 | **86.9** | 85.3 |
| LSUN_CR | **91.2** | 88.5 |
| Imagenet- | **86.5** | 84.8 |
| Noise | **94.8** | 91.2 |
| Uniform | **100.0** | **100.0** |

| CIFAR100 | ACET | ACET2 |
|---|---|---|
| SVHN | 73.9 | **84.6** |
| CIFAR10 | **77.2** | 77.0 |
| LSUN_CR | 78.0 | **80.0** |
| Imagenet- | **79.5** | 79.4 |
| Noise | 62.9 | **66.3** |
| Uniform | **100.0** | **100.0** |

Table 4: OOD detection performance (AUC in percent) for ACET (trained around tiny images) and ACET2 (trained around uniform noise).

| | | ACET | ACET2 |
|---|---|---|---|
| MNIST | TE | 0.6 | 0.6 |
| | SR | **0.0** | **0.0** |
| | AUC | **100.0** | **100.0** |
| FMNIST | TE | 4.8 | 4.6 |
| | SR | **0.0** | **0.0** |
| | AUC | **100.0** | **100.0** |
| SVHN | TE | 3.2 | 3.0 |
| | SR | **3.0** | 95.5 |
| | AUC | **96.5** | 5.4 |
| CIFAR10 | TE | 6.1 | 7.1 |
| | SR | **0.0** | 64.5 |
| | AUC | **99.9** | 35.9 |
| CIFAR100 | TE | 25.2 | 26.2 |
| | SR | **3.5** | 96.5 |
| | AUC | **95.8** | 14.6 |

Table 5: OOD detection performance (test error, AUC and success rate in percent) for ACET (trained around tiny images) and ACET2 (trained around uniform noise).

|  |  | Base | MCD | EDL | DE | GAN | ODIN | Maha | ACET | OE | CCU |
|---|---|---|---|---|---|---|---|---|---|---|---|
| **MNIST** | FMNIST | 97.5 | 89.4 | 99.4 | 99.4 | 99.4 | 98.8 | 97.0 | **100.0** | 99.9 | 99.9 |
|  | EMNIST | 77.9 | 60.0 | 77.3 | 84.5 | 85.5 | 78.4 | 74.4 | 90.9 | **91.4** | 84.3 |
|  | GrCIFAR10 | 99.7 | 91.1 | 99.8 | **100.0** | 99.5 | 99.9 | 98.9 | **100.0** | **100.0** | **100.0** |
|  | Noise | **100.0** | 75.5 | 99.8 | **100.0** | 99.2 | **100.0** | 96.5 | **100.0** | **100.0** | **100.0** |
|  | Uniform | 97.2 | 82.8 | 99.9 | 98.8 | 99.9 | 98.9 | **100.0** | **100.0** | **100.0** | **100.0** |
| **FMNIST** | MNIST | 97.6 | 79.3 | 95.9 | 97.5 | **99.9** | 99.2 | 97.2 | 97.4 | 97.0 | 98.3 |
|  | EMNIST | 96.8 | 74.2 | 94.6 | 96.1 | **100.0** | 98.9 | 96.5 | 97.0 | 98.6 | 99.1 |
|  | GrCIFAR10 | 92.2 | 92.7 | 86.9 | 90.8 | 82.3 | 92.9 | 98.6 | 96.8 | **100.0** | **100.0** |
|  | Noise | 93.8 | 78.6 | 91.6 | 92.5 | 95.4 | 95.4 | 97.0 | 95.3 | **100.0** | **100.0** |
|  | Uniform | 97.8 | 93.0 | 97.1 | 98.8 | 95.4 | 99.1 | 99.4 | **100.0** | 98.2 | **100.0** |
| **SVHN** | CIFAR10 | 97.2 | 96.2 | 98.5 | 99.2 | 98.6 | 97.3 | 99.0 | 97.3 | **100.0** | **100.0** |
|  | CIFAR100 | 96.7 | 95.8 | 98.3 | 99.0 | 98.2 | 96.6 | 98.8 | 97.0 | **100.0** | **100.0** |
|  | LSUN_CR | 99.9 | 99.9 | 99.9 | **100.0** | **100.0** | 99.9 | **100.0** | **100.0** | **100.0** | **100.0** |
|  | Imagenet- | 96.8 | 96.2 | 98.3 | 99.1 | 98.9 | 96.9 | 98.9 | 98.3 | **100.0** | **100.0** |
|  | Noise | 89.1 | 83.6 | 95.6 | 96.8 | 94.5 | 50.3 | **97.0** | 87.3 | 95.6 | 93.3 |
|  | Uniform | 98.5 | 97.0 | 98.7 | 98.5 | **100.0** | 98.9 | 99.3 | **100.0** | **100.0** | **100.0** |
| **CIFAR10** | SVHN | 92.3 | 71.1 | 90.7 | 87.4 | 80.5 | 92.7 | 85.9 | 91.0 | **98.5** | 97.5 |
|  | CIFAR100 | 86.3 | 80.0 | 89.3 | 89.8 | 84.0 | 85.5 | 83.4 | 87.4 | **95.6** | 94.6 |
|  | LSUN_CR | 99.7 | 99.2 | 99.7 | 99.7 | 99.7 | 99.7 | 99.6 | 99.7 | **100.0** | 99.9 |
|  | Imagenet- | 84.6 | 79.2 | 89.8 | 88.6 | 84.9 | 84.2 | 84.4 | 85.4 | **94.8** | 93.2 |
|  | Noise | 88.1 | 55.7 | 70.4 | 82.7 | 68.6 | 87.7 | 84.9 | 84.0 | 93.8 | **94.7** |
|  | Uniform | 97.8 | 83.4 | 94.5 | 98.0 | 82.4 | 99.1 | **100.0** | **100.0** | 99.2 | **100.0** |
| **CIFAR100** | SVHN | 67.4 | 52.9 | 72.0 | 75.6 | 63.4 | 71.0 | 69.1 | 59.4 | 89.6 | **90.8** |
|  | CIFAR10 | 80.9 | 66.1 | 75.8 | 79.1 | 72.5 | 81.3 | 61.2 | 79.7 | **84.8** | 83.7 |
|  | LSUN_CR | 99.3 | 98.3 | 99.0 | 99.3 | 99.2 | 99.3 | 99.2 | 99.1 | **99.9** | **99.9** |
|  | Imagenet- | 81.9 | 67.1 | 78.8 | 80.7 | 76.5 | 82.0 | 73.8 | 80.8 | **85.3** | 83.3 |
|  | Noise | 51.0 | 36.1 | 34.1 | 47.8 | 44.2 | 56.4 | 79.5 | 25.9 | 69.5 | **88.9** |
|  | Uniform | 95.6 | 66.1 | 53.1 | 60.0 | **100.0** | 96.0 | 96.4 | **100.0** | 99.5 | **100.0** |

Table 6: AUPR (in- versus out-distribution detection based on confidence/score) in percent for different OOD methods and datasets (higher is better). OE and CCU have the best OOD performance.

number of true positives (tp) over the sum of true positives and false positives (fp), i.e.

$$\text{precision} = \frac{\text{tp}}{\text{tp} + \text{fp}}. \tag{26}$$

Recall is defined as

$$\text{recall} = \frac{\text{tp}}{\text{tp} + \text{fn}}, \tag{27}$$

where fn is the number of false negatives. We report the AUPR for all models and all datasets in Table 6. Qualitatively we find that the results do not differ from the ones reported in Table 2.

