# OpenReview forum: "Towards neural networks that provably know when they don't know"
_ICLR.cc/2020/Conference — Accept (Poster)_

### Official Review · AnonReviewer1 · 2019-10-20
**Official Blind Review #1**

**Rating:** 6

**Review:**

The authors present a novel approach for OOD detection; in particular their approach comes with worst-case guarantees without compromising on performance.

The manuscript is clearly written and I only have some concerns regarding the evaluation.
First, while the authors include with MCD an uncertainty-aware training approach, I miss more state-of-the-art methods with substantially better OOD performance, including Evidential Deep Learning (Sensoy et al, NeurIPS 2018) and Deep Ensembles. In particular a comparison to EDL would be interesting, since a similar entropy-encouraging term in the loss function is used during training, resulting is maximum entropy for OOD samples.
Second, I would have liked to see precision and recall of the OOD detection task in addition to AUC, allowing a more meaningful/complete comparison between the approaches.

**Experience Assessment:**

I have published one or two papers in this area.

**Review Assessment: Checking Correctness Of Derivations And Theory:**

I carefully checked the derivations and theory.

**Review Assessment: Checking Correctness Of Experiments:**

I carefully checked the experiments.

**Review Assessment: Thoroughness In Paper Reading:**

I read the paper thoroughly.

---

> ### Author Response · Authors · 2019-11-13
> **Reply to Review #1**
>
> We thank the reviewer for suggesting the additional papers ``Evidential Deep Learning’’ (EDL) and ``Deep Ensembles’’ (DE) for comparison. It is fair to say that ``Deep Ensembles’’ has not been proposed for OOD detection but for uncertainty evaluation similar to MC-Dropout. We have included EDL and DE in our evaluation (please see Table 1 and 2 in the main paper). As pointed out by the reviewer, both techniques outperform MC-Dropout. DE performs better than EDL regarding OOD detection but both methods are not competitive with outlier exposure and our CCU – this holds for all datasets but with particularly large differences on SVHN, CIFAR10 and CIFAR100 . EDL and DE fail regarding the worst case evaluation on adversarial noise. For EDL the AUC is zero on all datasets, meaning that for all 200 certified balls for each dataset the maximal confidence achieved in the ball around uniform noise is higher than the confidence of all test samples. Likewise DE has an AUC of below 8% on all datasets in this worst-case setting.
>
> ``I would have liked to see precision and recall of the OOD detection task in addition to AUC’’
>
> In Appendix G we have added a table with the AUPR, the area under the precision-recall curve,
> as discussed in Hendrycks, Gimpel ``A baseline for detecting misclassified and out-of-distribution examples in neural networks’’, ICLR 2017. We did not add individual ROC curves or precision-recall curves as this would result in 28 plots each with 10 curves (note that we compare 10 different methods) which becomes quite cluttered. However, if the reviewers and/or the area chair indicate that they would like to see these plots, we are happy to include them in the final version.

---

> > ### Comment · AnonReviewer1 · 2019-11-14
> > **Reply  to authors**
> >
> > The authors have addressed my mainn concerns in full and I will update my score accordingly.

---

### Official Review · AnonReviewer2 · 2019-10-21
**Official Blind Review #2**

**Rating:** 6

**Review:**

Summary: This paper provides a method to train neural networks with guarantees on outputting probabilities/scores close to uniform on inputs that are out of domain.
Precisely, on some point x_0, we can obtain the largest radius such that on inputs in the ball around x_0 the classifier outputs some predetermined  maximum score.
The paper combines a simple generative model (mixture of Gaussian) for modeling in-distribution vs. out of distribution. The simplicity allows to obtain guarantees on the probability of an input being considered out of class. The model output p(y | x) critically uses the probability of being out of distribution. Hence guarantees on being detected as out of distribution translate to guarantees on p(y|x) being close to uniform.

Decision: I vote for accepting the paper. The paper has some clear strengths. However, I have some concerns regarding experiments and comparison to previous work. It would be great if the authors could clarify.

Strengths: The paper’s methodolgy is clearly written.  The modeling is clear and sound. Overall, this idea is a promising approach to obtain networks that are provably under-confident far from training examples. The training cost for this approach is comparable to standard training, and the approach seems scalable and broadly applicable in general.

The experimental evaluation is also clearly described. Worst-case evaluation of OOD (out of domain) performance seems novel and the gains not this objective using the proposed approach of this paper are interesting and promising.

Concerns: While the paper explains the proposed method well, the description of previous work and relation to previous work is inadequate. After spending some hours reading the cited paper, I am still confused about what’s the novelty of this work at a high level.
This work uses p(x|i) and p(x|o) in the computation of p(y|x) during inference, where i and o are in distribution and out of distribution respectively. This is crucial to obtain guarantees one performance. However, how does this compare to other previous work that also uses some kind of generative modeling to model in/out distribution? A bunch of papers are cited in the introduction as doing this, but the relationship to the proposed work is unclear.

— I have a major experimental concern: When comparing against ACET, the baseline of performing adversarial style training on random noise inputs seems more appropriate since it’s closer to the evaluation metric (which picks random noise as out of domain and not 80M Tiny Images). What does the performance of this ACET baseline look like?

— During evaluation, how do you ensure that the radius is not too large such that it has images that the model should actually be confident on (close to in distribution samples). A flip-side is how sensitive the performance is to the score chosen for choosing the radius for computing the valid set of images for the adversary? It’s possible that the 11%  threshold is too high? What’s the minimum confidence of CCU on the test images?

Minor comments on writing:
—K_i, K_0 are not defined when writing the expression for GMM


**Experience Assessment:**

I have read many papers in this area.

**Review Assessment: Checking Correctness Of Derivations And Theory:**

I assessed the sensibility of the derivations and theory.

**Review Assessment: Checking Correctness Of Experiments:**

I assessed the sensibility of the experiments.

**Review Assessment: Thoroughness In Paper Reading:**

I read the paper at least twice and used my best judgement in assessing the paper.

---

> ### Author Response · Authors · 2019-11-13
> **Reply to Review #2**
>
> Thanks a lot for the encouraging comments and for the interesting questions. We address all mentioned concerns:
>
> ``What is the novelty of the method at a high level?’’
>
> We are not aware that any other paper on OOD detection has written it in a Bayesian framework as we do modeling explicitly p(x|i) and p(x|o) and thus having an expression of p(y|x) in terms of p(y|x,i) and p(y|x,o). This then allows us to derive our CCU optimization framework as maximum likelihood estimation in this particular model. Most other approaches are more ad-hoc by enforcing e.g. low confidence on the out-distribution data. However, the main novelty compared to any other OOD method is that our approach allows to certify whole volumes to have low confidence and thus being able to certify that this volume will be identified by the classifier as ``out-distribution”. This is done by using a density estimator for p(x|i) and p(x|o) which one can control (this is the reason for the ``simple’’ Gaussian mixture models for p(x|i) and p(x|o)) . Moreover, we bring up the challenging worst case evaluation of adversarial noise which we think should become standard in OOD detection. For the use in safety-critical systems the average case in terms of empirical evaluation on ``out-of-distribution’’ datasets is in our opinion not sufficient. Thus, we believe that our provable guarantees for the certification of low confidence over whole volumes and provably low confidence far away from the training data constitute important steps towards a ``general certification’’ of neural networks.
> We have changed the introduction to highlight our contributions more clearly.
>
>
> ``What is the performance of ACET when trained on adversarial uniform noise instead of adversarial tiny image dataset”
>
> We have added this comparison in Appendix F. In fact we did in the beginning train ACET using adversarial noise as in the original paper but we found out that the performance of ACET on OOD detection improves when training on adversarial tiny images. Moreover, we wanted to have all methods to be trained using the same information on the out-distribution so that one has a fair comparison. In short the results in Appendix F show that OOD detection of the ACET trained on
> adversarial tiny images is better for OOD detection but even for the worst case evaluation on adversarial noise. The reason is that the attack model is an l_2-ball with respect to a metric which is adapted to ``natural images’’ (we use the covariance of the training data) and not the l_infty attack
> which ACET does during training. As our ACET model is trained using tiny images it is more adapted to these natural images.
>
>
> ``How do you ensure that the radius is not too large such that it has images that the model should actually be confident on (close to in distribution samples)?’’
>
> Thanks for this question – we provide a detailed analysis in Appendix E. Indeed for all our 200 certified balls for each dataset we checked if they contain images from the training or test set and this does not happen on any of the datasets. Thus we think that our certification procedure works very reliably.
>
>
> ``A flip-side is how sensitive the performance is to the score chosen for choosing the radius for computing the valid set of images for the adversary? It’s possible that the 11% threshold is too high? What’s the minimum confidence of CCU on the test images?’’
> We also have analyzed this in Appendix E. For MNIST, FMNIST and SVHN no test images resp. a tiny fraction (less than 0.1%) gets less than 11% confidence. For Cifar100 1.3% of the test images get less than 1.1% confidence, and for Cifar10 5.3% less than 11%.
> However, one has to note that the certified bound of 11% confidence is an upper bound which is not tight in practical settings. Thus the actual maximal confidence over all the 200 certified balls of CCU which we found using PGD was never larger than the minimal confidence achieved on the test set (this is why CCU always has 100% AUC). However, in order to address the possibility that this could be an artifact of PGD not finding the global maximum on our certified balls, now we also report a lower bound on the AUC assuming that the maximal confidence of 11% (resp. 1.1%) would be attained by CCU in all the 200 certified balls. Note that even this theoretical lower bound still outperforms the other models’ empirical AUC.

---

### Official Review · AnonReviewer3 · 2019-10-23
**Official Blind Review #3**

**Rating:** 6

**Review:**

This paper uses a generative model to assign anomaly scores. By its construction, it can provide provable performance guarantees. Experiments do not make unreasonable assumptions such as the ability to peak at the test data, unlike much previous work.
My primary concern is that they should show performance on CIFAR-100 not just CIFAR-10, and I certainly hope these experiments will be included during the rebuttal. Overall experimentation is thorough and competently executed, and the proposed technique is sufficiently novel.

Small comments:

> adv OOD detection with uniform ball perturbed
This is a good way of formulating adversarial OOD detection.

A possibly related work is _Early Methods for Detecting Adversarial Images_ (2016) since it uses covariance matrix information for detecting adversarial examples. This paper should cite _Open Category Detection with PAC Guarantees_ by Liu et al. (ICML 2018) since this also involved provable guarantees for OOD detection.

Update: my concerns are addressed but my sentiment is still that this is a 6.

**Experience Assessment:**

I have published in this field for several years.

**Review Assessment: Checking Correctness Of Derivations And Theory:**

I assessed the sensibility of the derivations and theory.

**Review Assessment: Checking Correctness Of Experiments:**

I carefully checked the experiments.

**Review Assessment: Thoroughness In Paper Reading:**

I read the paper at least twice and used my best judgement in assessing the paper.

---

> ### Author Response · Authors · 2019-11-13
> **Reply to Review #3**
>
> We appreciate the helpful feedback from the reviewer and for pointing out relevant references. We now discuss these references in the related work in the introduction.
>
> The suggested paper by Liu et al, “Open Category Detection with PAC Guarantees” also yields guarantees for OOD detection but of a completely different kind than discussed in our paper. They provide  guarantees on the generalization of the employed empirical performance measures, while we certify low confidence of a classifier on whole volume. We are thankful for bringing up
> Hendrycks, Gimpel “Early methods for detecting adversarial samples” as they have employed a similar metric to the one we use in our Gaussian mixture model (which we now cite there).
>
> As requested we have now included CIFAR100 in our evaluation (please see our enlarged result tables in the paper). The results are similar to the other datasets, in the sense that Outlier exposure and our CCU are the best methods for OOD detection and CCU again succeeds in the worst case scenario of an attack on an OOD system with adversarial noise. All other methods except ACET fail with AUC values below 20%.
>
> Please note that due to a change in Pytorch we had to retrain all models in order to run additional analysis. Thus there are slight changes in the reported numbers for all methods but there are no changes in the overall picture.

---

### Decision · Program_Chairs · 2019-12-19

**Decision:**

Accept (Poster)

**Comment:**

This paper tackles the problem of confidence on neural network predictions for out-of-distribution (OOD) samples. The authors propose an approach for training neural networks such that the OOD prediction is uniform across classes. The approach requires samples from in- and out-of distribution and relies on a mixture of Gaussians for modelling the distributions, allowing to obtain theoretical guarantees on detecting OOD samples (unlike existing techniques).

The main concerns of the reviewers have been addressed during the rebuttal. If this approach does not outperform state-of-the-art in practice, providing such theoretical guarantees is an important contribution.

All reviewers agree that this paper should be accepted. I therefore recommend acceptance.